# Assessment of the Relationship Between Positive Radial Margin and Prognosis in Patients with Gastric Adenocarcinoma

**DOI:** 10.3390/cancers17091463

**Published:** 2025-04-27

**Authors:** Yu-Chih Wang, Yi-Ju Chen, Yu-Hsuan Shih, Feng-Hsu Wu

**Affiliations:** 1Division of General Surgery, Department of Surgery, Taichung Veterans General Hospital, Taichung 407219, Taiwan; rail127.127@gmail.com (Y.-C.W.); chenyiju5668@gmail.com (Y.-J.C.); 2Division of Medical Oncology, Department of Oncology, Taichung Veterans General Hospital, Taichung 407219, Taiwan; rollingstone07@gmail.com; 3Department of Post-Baccalaureate Medicine, College of Medicine, National Chung Hsing University, Taichung 402202, Taiwan; 4Department of Critical Care Medicine, Taichung Veterans General Hospital, Taichung 407219, Taiwan

**Keywords:** adenocarcinoma, gastrectomy, margins of excision, prognosis

## Abstract

This study assesses the influence of a positive radial margin on the outcomes of gastric adenocarcinoma patients post-gastrectomy. A cohort of 431 stage II/III patients operated on from 2009 to 2019 was retrospectively examined. Of these, 94 patients (21.8%) presented with a positive margin, with radial margin positivity at 16.2%. Positive margins were correlated with perineural invasion and advanced cancer stages. Advanced Borrmann type, positive nodal involvement, higher nodal burden (≥5), and margin status were associated with decreased overall survival (OS) and disease-free survival (DFS). Subgroup analysis indicated that radial margin positivity notably impacted OS and DFS in patients with advanced T stage and nodal involvement. These results imply that aggressive tumor biology may lead to margin positivity, underscoring the necessity for personalized treatment approaches for patients exhibiting a positive radial margin.

## 1. Introduction

Gastric cancer ranks as the fifth most prevalent malignant tumor globally, with its mortality rate standing fourth among all malignancies. In 2020, nearly two-thirds of gastric cancer diagnoses occurred in Eastern and Southeastern Asia [1]. Margin-negative resection with appropriate lymphadenectomy is considered to be a mandatory component of potentially curative treatment in patients with stage II/III locally invasive gastric cancer according to the Japanese Gastric Cancer Treatment Guidelines 6th edition [2].

Surgical margin status is a crucial determinant of survival outcomes in gastric cancer patients, with numerous studies showing that positive proximal and distal resection margins are associated with poor prognoses and with one study reporting that microscopically positive resection margins occur in 2.3% to 11.2% of patients with potentially resectable gastric cancer [3,4,5,6].

However, current research on the impact of a positive radial margin remains limited. According to the College of American Pathologists (CAP) protocol [7] for reporting gastric cancer pathology, radial margins are a significant factor to assess. The prognostic impact of a positive radial margin remains unclear.

At our institution, pathologists follow the CAP guidelines for gastric cancer pathology, including radial margin assessment, identifying some patients as microscopically positive for radial margins, and this study aims to explore the influence of positive radial margins on overall survival (OS) and disease-free survival (DFS) in patients with gastric adenocarcinoma.

## 2. Materials and Methods

### 2.1. Patients

This was a retrospective cohort study of patients with pathological stage II or III gastric adenocarcinoma who underwent gastrectomy at Taichung Veterans General Hospital. Inclusion criteria were adults with gastric adenocarcinoma who received curative-intent gastrectomy and lymph node dissection at the institution between January 2009 and December 2019. Pathology reporting and staging followed the CAP protocol for gastric cancer.

### 2.2. Statistical Analyses

Statistical analyses were performed to examine factors associated with positive margin status, as well as the impact of margin status on OS and DFS. Logistic regression was used to evaluate clinicopathologic variables predicting positive margin involvement. Cox’s proportional hazards regression assessed variables affecting OS and DFS. The log-rank test was employed to compare OS and DFS curves stratified by margin status, including subgroup analyses of patients with advanced T3/T4 tumors and those with node-positive disease.

Appropriate univariable and multivariable analyses were conducted, with results reported as odds ratios (ORs), hazard ratios (HRs), 95% confidence intervals (CIs), and *p*-values where applicable. *p*-values < 0.05 were considered statistically significant. Analyses were performed using the Statistical Package for the Social Science (IBM SPSS version 22.0; International Business Machines Corp, New York, NY, USA). The institutional review board (IRB) of Taichung Veterans General Hospital approved the present study (IRB number: CE24273B).

### 2.3. Radial Margin

Radial margins refer to the nonperitonealized soft tissue margins nearest to the deepest tumor penetration. In the stomach, the radial margins consist solely of the lesser omental (hepatoduodenal and hepatogastric ligaments) and greater omental resection margins and the circumference margins of stomach specimens [7].

### 2.4. Gastrectomy

The policy of performing gastrectomy followed the Japanese Gastric Cancer Treatment Guidelines 6th edition [2]. Intraoperative frozen sections on distal margin or proximal margin were sent to a pathologist for analysis by the different surgeons’ preferences. If a frozen section was positive on the proximal or distal margin, extensive resection was performed to ensure a negative proximal or distal margin. However, frozen sections were not performed on radial margin analysis during operation.

## 3. Results

### 3.1. Patients Characteristics

Table 1 compares the characteristics of the 431 gastric cancer patients who underwent gastrectomy with either a positive (*n* = 94) or negative (*n* = 337) margin status. The groups did not differ significantly in terms of sex, age, receipt of adjuvant chemotherapy, tumor differentiation, signet ring cell type, surgery type, or extent of lymph node dissection. However, patients with positive margins had a lower body mass index (BMI), a more advanced stage, perineural invasion, diffuse Borrmann types, deeper tumor invasion, and a higher nodal burden compared to those with negative margins. In terms of margin type, 0 means negative for a proximal, distal and radial margin, while ‘’other’’ includes a mixed positive radial margin and positive proximal or distal margin and merely a positive proximal or distal margin. The number of mixed positive radial margins and either positive proximal or distal margins was 10. The number of merely positive proximal or distal margins was 14. Therefore, the total number in “other” was 24.

### 3.2. Factors Related with a Positive Margin

Table 2 presents the results of the univariate and multivariable logistic regression analyses examining the risk factors associated with positive margin status. In the univariate analysis, lower BMI (OR 0.90, 95% CI 0.84–0.96, *p* = 0.003), advanced stage III disease (OR 3.48, 95% CI 2.02–6.02, *p* < 0.001), perineural invasion (OR 4.37, 95% CI 2.45–7.81, *p* < 0.001), Borrmann type 3 (OR 6.59, 95% CI 1.55–28.02, *p* = 0.01) or type 4 (OR 9.25, 95% CI 1.71–50.01, *p* = 0.01), and having more than five positive lymph nodes (OR 2.20, 95% CI 1.38–3.50, *p* = 0.001) were significantly associated with increased odds of positive margin status. In the multivariable analysis adjusting for other factors, lower BMI (OR 0.90, 95% CI 0.84–0.97, *p* = 0.005), stage III disease (OR 2.06, 95% CI 1.05–4.03, *p* = 0.04), and perineural invasion (OR 3.29, 95% CI 1.75–6.17, *p* < 0.001) remained independent predictors of positive margin status.

### 3.3. Factors Related with Overall Survival

Table 3 reports the results of variables that could potentially impact overall survival in our patient group. In the multivariable analysis adjusting for other factors, adjuvant chemotherapy use (HR 0.27, 95% CI 0.17–0.45, *p* < 0.001) and D2 or greater lymph node dissection (HR 0.59, 95% CI 0.38–0.91, *p* = 0.02) were independently associated with improved survival. Conversely, Borrmann type 3 (HR 3.33, 95% CI 1.14–9.73, *p* = 0.028) or type 4 (HR 4.60, 95% CI 1.23–17.20, *p* = 0.02), node positivity (HR 4.64, 95% CI 1.88–11.42, *p* = 0.001), higher nodal burden > 5 nodes (HR 1.90, 95% CI 1.24–2.93, *p* = 0.003), and positive margin status (HR 1.91, 95% CI 1.30–2.82, *p* = 0.001) were independently predictive of worse overall survival.

### 3.4. Outcome Analysis: Overall Survival and Disease-Free Survival

Table 4 presents the analysis of disease-free survival. For the disease-free survival analysis, the multivariable analysis showed that adjuvant chemotherapy (HR 0.31, 95% CI 0.21–0.45, *p* < 0.001), D2 lymph node dissection (HR 0.70, 95% CI 0.50–0.99, *p* = 0.04), and lower BMI (HR 0.96, 95% CI 0.92–1.00, *p* = 0.04) were independently associated with improved disease-free survival. Conversely, advanced stage III (HR 2.11, 95% CI 1.27–3.49, *p* = 0.004), Borrmann type 3 (HR 2.59, 95% CI 1.20–5.60, *p* = 0.02) and type 4 (HR 4.22, 95% CI 1.56–11.41, *p* = 0.005), node positivity (HR 2.37 for N1, 95% CI 1.25–4.50, *p* = 0.008), higher nodal burden > 5 nodes (HR 1.68, 95% CI 1.21–2.35, *p* = 0.002), and positive margin status (HR 1.45, 95% CI 1.05–1.99, *p* = 0.02) were independently predictive of worse disease-free survival.

### 3.5. Subgroup Analysis: Relationship Between Radial Margin and Overall Survival and Disease-Free Survival in Advanced T Stage

Figure 1 presents the log-rank test analysis comparing OS and DFS among patients with T3 or T4 tumors based on their margin status. The 5-year OS rates were 63.7% for patients with negative margins, 47.4% for those with positive radial margins, and 29.8% for other involved margins (*p* < 0.001 by log-rank test). In pairwise comparisons using the log-rank test, a positive radial margin was associated with significantly worse OS compared to a negative margin (*p* = 0.003). For DFS, the 5-year rates were 47.9% with negative margins, 26.5% with positive radial margins, and 15.6% with other involved margins (*p* < 0.001). Similarly, pairwise log-rank tests showed that a positive radial margin predicted worse DFS than a negative margin status (*p* = 0.003). The Kaplan–Meier curves visually depict the separation in OS and DFS based on margin status.

### 3.6. Subgroup Analysis: Relationship Between Radial Margin and Overall Survival and Disease-Free Survival in Nodal Positive Disease

Figure 2 presents the analysis of the node-positive subgroup. The 5-year OS rates were 63.8% for patients with negative margins, 44.3% for those with positive radial margins, and 26.0% for other involved margins (*p* < 0.001 by log-rank test). In pairwise comparisons, a positive radial margin was associated with significantly worse OS compared to a negative margin (*p* < 0.001), while other involved margins also predicted poorer OS than negative margins (*p* < 0.001) but were not significantly different from positive radial margins (*p* = 0.24). Regarding DFS, the 5-year rates were 49.9% with negative margins, 20.5% with positive radial margins, and 16.6% with other involved margins (*p* < 0.001). Pairwise log-rank tests showed that both positive radial (*p* < 0.001) and other involved margins (*p* < 0.001) had significantly worse DFS compared to negative margin status in node-positive patients. Kaplan–Meier curves are presented below.

## 4. Discussion

This retrospective cohort study at a single institute aimed to determine whether a positive radial margin in patients receiving gastrectomy for gastric adenocarcinoma impacts OS and DFS. Additionally, we analyzed the risk factors leading to potential positive margin status and the factors that might affect survival. We found that advanced stage and perineural invasion were potential risk factors for a positive margin, with a more advanced Borrmann type and higher lymph node burden leading to a tendency for positive margin status. For OS, advanced Borrmann type, positive nodal status, nodal burden greater than five nodes, and positive margin status were associated with poorer overall survival. For DFS, all the above factors plus advanced stage were recognized to shorten DFS. In the subgroup analysis, radial margin positivity negatively affected OS and DFS in the advanced-T-stage group and the node-positive group.

The positive margin rate in this cohort study was 21.8%, with radial margin positivity accounting for 16.2%. In previous studies, the margin status, primarily the proximal and distal margin status, ranged widely from 0.8% to 23% [3,8]. No study was found about the prevalence of positive radial margins.

For factors linked to positive surgical margins in individuals diagnosed with gastric cancer, Juez et al. [9] stated that advanced tumor stage, T stage, N stage, median number of positive nodes, diffuse Lauren type, whole stomach involvement, and poorly differentiated tumors were associated with positive margin status. In Liang et al.’s study [10], infiltrative growth patterns (Borrmann type III and IV) and extranodal metastasis were found to be independent risk factors for positive margins. Tu et al. [11] also demonstrated that stage T3-4, lymph node metastasis, and M1 disease were relevant to positive margin status. All of these parameters are related to the aggressiveness of the tumor. The higher the aggressiveness, the more likely a positive margin would be observed after gastrectomy. Therefore, if such parameters are encountered, wider excision should be considered to achieve R0 resection. In our study, we found that a more advanced tumor stage (stage III disease versus stage II disease) and positive perineural invasion were factors related to positive margins. Additionally, lymphovascular invasion, advanced Borrmann type, and higher nodal burden were associated with a tendency for positive margins. This correlates with previous studies, indicating the relationship between positive margins and tumor aggressiveness.

Regarding factors related to OS and DFS, there are still debates about whether positive margin status, primarily proximal and distal margin status, negatively affects OS and DFS. Reviewing the literature [12,13,14,15], positive margin status was reported to negatively impact OS and DFS. However, in Kim et al.’s study [16], they demonstrated that the negative impact of positive margins on OS is limited to cases where the positive node count is five or fewer. When the count of positive nodes surpasses five, the tumor’s aggressiveness outweighs the influence of positive margin status. According to Sun et al. [17], substantial disparities in survival were evident between patients with negative and positive margins among those with pT1-2, pN0-1, and stages I–II receiving D2/D3 lymphadenectomy. However, such differences were not apparent among those with pT3-4, pN2-3, and stages III–IV. The diminishing prognostic relevance suggests that the notably poorer prognosis observed in patients with positive margins might be attributed, in part, to their more frequent diagnosis at an advanced stage compared to patients with negative margins. The results of our study were in accordance with previous studies, indicating that positive margin status is one of the worst prognostic factors for OS and DFS. Moreover, we also presented subgroup analyses, proving that positive margin status negatively affects OS and DFS.

The focal aspect of our study revolves around determining whether radial margin status could influence both OS and DFS. In the subgroup analysis, regardless of advanced T stage status or nodal positivity, radial margin positivity negatively affected OS and DFS under the log-rank test. This finding underscores the importance of considering radial margin status in pathological reports. According to Ma et al. [18], 69 patients with gastric or esophageal cancer underwent surgery and were found to have R1 resections. Among them, some had positive radial margins, while others had positive proximal or distal margins or mixed positive margins. In patients with R1 resections, the pathological stage was more advanced, and the prognosis was relatively poor. To our knowledge, radial margin status is not reported in pathology in several hospitals in Taiwan. Missing this important factor in pathological reports could lead to inadequate prognosis evaluation and deficient treatment options. In Stiekema et al.’s study [19], R1 resection did not emerge as a detrimental prognostic factor in gastric cancer patients who received chemoradiation therapy (CRT) following the surgical procedure. This suggests that in our patient group, if radial margin positivity is noted, CRT could be offered as a treatment option. Further studies are required to confirm this hypothesis.

The limitation of this study is the small sample size due to it being a retrospective cohort study of a single medical center in Taiwan. The patients enrolled in the study ranged from 2009 to 2019. Throughout this period, alterations in the clinical management of gastric adenocarcinoma and surgical techniques may have improved, potentially impacting our findings. Additionally, this study was conducted in a medical center where proficient surgeons performed all gastrectomies in partnership with specialized gastrointestinal pathologists. Consequently, our findings might not be universally applicable to other facilities with distinct treatment approaches and varying levels of surgical proficiency.

## 5. Conclusions

Worse overall survival and disease-free survival were linked to positive radial margin status. Microscopic positivity at radial margins following curative-intent gastrectomy was linked to worse outcomes, including overall survival and disease-free survival. To improve prognosis, R1 resection should be avoided, and the use of intraoperative frozen section analysis may help verify margin status during surgery. Detailed descriptions in pathological reports are required, and future treatment plans for patients with positive radial margins should be further explored.

## Figures and Tables

**Figure 1 cancers-17-01463-f001:**
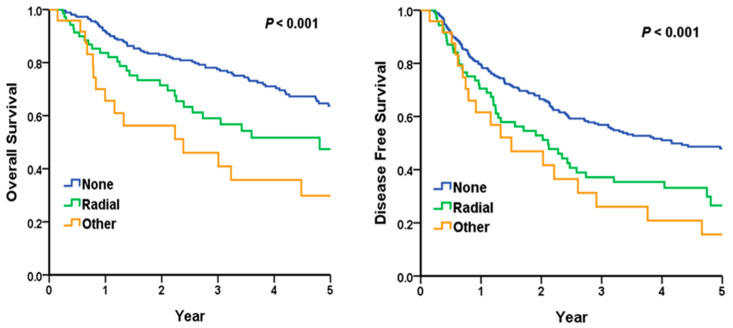
Subgroup analysis: the relationship between radial margin and overall survival (**left**) and disease-free survival (**right**) in advanced T stage. The relationship between radial margin and overall survival (**left**) and disease-free survival (**right**) in advanced T stage. This analysis focused on patients with advanced T stages (T3 and T4). The impact of radial margin status on overall survival and disease-free survival was assessed. Patients with positive radial margins exhibited significantly worse overall survival (*p* = 0.003) and disease-free survival (*p* = 0.003) compared to those with negative margins.

**Figure 2 cancers-17-01463-f002:**
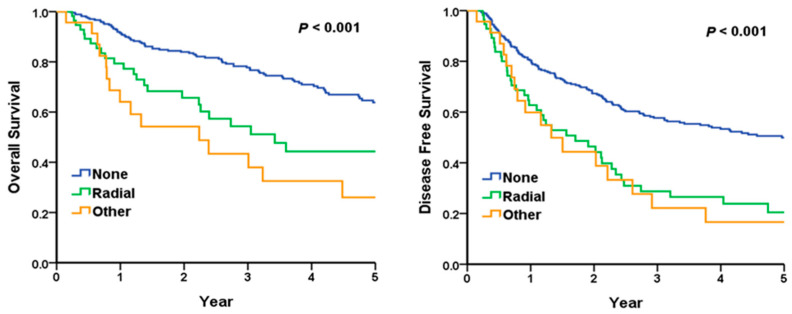
Subgroup analysis: the relationship between radial margin and overall survival (**left**) and disease-free survival (**right**) in nodal positive status. The relationship between radial margin and overall survival (**left**) and disease-free survival (**right**) in nodal positive status. This analysis focused on patients with positive N stages (N1, N2 and N3). The impact of radial margin status on overall survival and disease-free survival was evaluated. Patients with positive radial margins demonstrated significantly worse overall survival (*p* < 0.001) and disease-free survival (*p* < 0.001) compared to those with negative margins.

**Table 1 cancers-17-01463-t001:** Patient characteristics.

	ma Negative		ma Positive		*p* Value
N	337		94		
Sex					0.643
Female	138	(40.9%)	36	(38.3%)	
Male	199	(59.1%)	58	(61.7%)	
Age	67.0	(56.0–77.0)	67.0	(57.3–78.0)	0.744
Adjuvant chemotherapy	266	(78.9%)	79	(84.0%)	0.273
BMI	23.3	(21.3–25.8)	22.1	(20.1–24.4)	0.002 **
Differentiation					0.150
Well to moderate	52	(15.4%)	9	(9.6%)	
Poor differentiation	285	(84.6%)	85	(90.4%)	
Signet ring feature	123	(36.7%)	43	(45.7%)	0.112
Gastrectomy type					0.864
Subtotal gastrectomy	229	(68.0%)	63	(67.0%)	
Total gastrectomy	108	(32.0%)	31	(33.0%)	
Lymphadenectomy type					0.407
<D2 dissection	86	(25.5%)	28	(29.8%)	
D2 dissection	251	(74.5%)	66	(70.2%)	
Stage					<0.001 **
Stage II	158	(46.9%)	19	(20.2%)	
Stage III	179	(53.1%)	75	(79.8%)	
Lymphovascular invasion	242	(72.0%)	74	(78.7%)	0.193
Perineural invasion	175	(52.7%)	78	(83.0%)	<0.001 **
site					0.196
Upper	72	(21.4%)	12	(12.8%)	
Middle	34	(10.1%)	15	(16.0%)	
Lower	188	(55.8%)	51	(54.3%)	
More than one site	25	(7.4%)	9	(9.6%)	
Stump	18	(5.3%)	7	(7.4%)	
Borrmann type					0.003 **
0	20	(5.9%)	0	(0.0%)	
1	17	(5.0%)	2	(2.1%)	
2	81	(24.0%)	12	(12.8%)	
3	205	(60.8%)	73	(77.7%)	
4	14	(4.2%)	7	(7.4%)	
T stage					0.006 **
1	22	(6.5%)	0	(0.0%)	
2–4	315	(93.5%)	94	(100.0%)	
N stage					0.426
0	66	(19.6%)	15	(16.0%)	
1–3	271	(80.4%)	79	(84.0%)	
Positive nodal number					0.001 **
≤5	219	(65.0%)	43	(45.7%)	
>5	118	(35.0%)	51	(54.3%)	
Margin type					<0.001 **
0	337	(100.0%)	0	(0.0%)	
Radial	0	(0.0%)	70	(74.5%)	
Other	0	(0.0%)	24	(25.5%)	

Chi-square test or Mann–Whitney’s U test. Median (IQR) ** *p* < 0.01.

**Table 2 cancers-17-01463-t002:** Variables related to positive margin status.

	Univariate				Multivariable			
	OR	95%CI		*p* Value	OR	95%CI		*p* Value
Sex								
Female	Reference							
Male	1.12	(0.70–1.79)		0.643				
Age	1.00	(0.99–1.02)		0.780				
Adjuvant chemotherapy	1.41	(0.76–2.59)		0.275				
BMI	0.90	(0.84–0.96)		0.003 **	0.90	(0.84–0.97)		0.005 **
Differentiation								
Well to moderate	Reference							
Poor differentiation	1.72	(0.82–3.64)		0.154				
Signet ring feature	1.45	(0.91–2.31)		0.113				
Gastrectomy type								
Subtotal gastrectomy	Reference							
Total gastrectomy	1.04	(0.64–1.70)		0.864				
Lymphadenectomy type								
<D2 dissection	Reference							
D2 dissection	0.81	(0.49–1.34)		0.407				
Stage								
Stage II	Reference				Reference			
Stage III	3.48	(2.02–6.02)		<0.001 **	2.06	(1.05–4.03)		0.036 *
Lymphovascular invasion	1.44	(0.83–2.49)		0.195				
Perineural invasion	4.37	(2.45–7.81)		<0.001 **	3.29	(1.75–6.17)		<0.001 **
site								
Upper	0.61	(0.31–1.22)		0.163				
Middle	1.63	(0.82–3.22)		0.162				
Lower	Reference							
More than one site	1.33	(0.58–3.02)		0.500				
Stump	1.43	(0.57–3.62)		0.446				
Borrmann type								
0–1	Reference				Reference			
2	2.74	(0.58–12.87)		0.201	1.24	(0.25–6.20)		0.791
3	6.59	(1.55–28.02)		0.011 *	2.40	(0.53–10.91)		0.258
4	9.25	(1.71–50.01)		0.010 *	2.25	(0.38–13.33)		0.373
T stage								
1	Reference							
2–4	--							
N stage								
0	Reference							
1–3	1.28	(0.69–2.37)		0.427				
Positive nodal number								
≤5	Reference				Reference			
>5	2.20	(1.38–3.50)		0.001 **	1.26	(0.72–2.23)		0.417

Logistic regression. * *p* < 0.05, ** *p* < 0.01.

**Table 3 cancers-17-01463-t003:** Variables related to overall survival after gastrectomy.

	Univariate				Multivariable			
	Hazard Ratio	95%CI		*p* Value	Hazard Ratio	95%CI		*p* Value
Sex								
Female	Reference				Reference			
Male	1.52	(1.05–2.18)		0.025 *	1.01	(0.68–1.50)		0.956
Age	1.02	(1.01–1.04)		0.003 **	1.01	(0.99–1.03)		0.206
Adjuvant chemotherapy	0.46	(0.31–0.70)		<0.001 **	0.27	(0.17–0.45)		<0.001 **
BMI	0.95	(0.91–1.00)		0.056				
Differentiation								
well to moderate	Reference							
poor differentiation	1.38	(0.79–2.40)		0.255				
Signet ring feature	1.03	(0.72–1.46)		0.879				
Gastrectomy type								
Subtotal gastrectomy	Reference				Reference			
Total gastrectomy	1.93	(1.36–2.73)		<0.001 **	1.19	(0.60–2.36)		0.627
Lymphadenectomy type								
<D2 dissection	Reference				Reference			
D2 dissection	0.60	(0.41–0.86)		0.006 **	0.59	(0.38–0.91)		0.016 *
Stage								
Stage II	Reference				Reference			
Stage III	3.81	(2.49–5.83)		<0.001 **	1.74	(0.92–3.30)		0.088
Lymphovascular invasion	1.73	(1.11–2.70)		0.015 *	0.58	(0.35–0.97)		0.037
Perineural invasion	1.96	(1.34–2.87)		0.001 **	1.36	(0.87–2.13)		0.174
site								
Upper	1.20	(0.75–1.94)		0.445	0.94	(0.43–2.07)		0.883
Middle	0.91	(0.48–1.73)		0.776	0.91	(0.45–1.82)		0.788
Lower	Reference				Reference			
more than one site	2.62	(1.57–4.38)		<0.001 **	1.85	(0.78–4.41)		0.165
stump	2.32	(1.22–4.41)		0.010 *	2.28	(0.90–5.81)		0.083
Borrmann type								
0–1	Reference				Reference			
2	1.39	(0.51–3.83)		0.522	1.50	(0.46–4.86)		0.498
3	3.62	(1.47–8.91)		0.005 **	3.33	(1.14–9.73)		0.028 *
4	9.27	(3.29–26.13)		<0.001 **	4.60	(1.23–17.20)		0.023 *
T stage								
1	Reference				Reference			
2–4	4.99	(1.23–20.19)		0.024*	1.92	(0.41–8.89)		0.406
N stage								
0	Reference				Reference			
1–3	4.17	(2.04–8.53)		<0.001 **	4.64	(1.88–11.42)		0.001 **
Positive nodal number								
≤5	Reference				Reference			
>5	2.91	(2.05–4.14)		<0.001 **	1.90	(1.24–2.93)		0.003 **
margin type								
0	Reference				Reference			
radial	2.32	(1.53–3.54)		<0.001 **				
other	3.86	(2.26–6.59)		<0.001 **				
margin								
Negative	Reference				Reference			
Positive	2.70	(1.88–3.88)		<0.001 **	1.91	(1.30–2.82)		0.001 **

Cox’s proportional hazard regression. * *p* < 0.05, ** *p* < 0.01.

**Table 4 cancers-17-01463-t004:** Variables related to disease-free survival after gastrectomy.

	Univariate				Multivariable			
	Hazard Ratio	95%CI		*p* Value	Hazard Ratio	95%CI		*p* Value
Sex								
Female	Reference				Reference			
Male	1.66	(1.24–2.22)		0.001 **	1.29	(0.94–1.78)		0.119
Age	1.02	(1.01–1.03)		0.001 **	1.01	(0.99–10.2)		0.316
Adjuvant chemotherapy	0.45	(0.33–0.62)		<0.001 **	0.31	(0.21–0.45)		<0.001 **
BMI	0.96	(0.92–1.00)		0.038 *	0.96	(0.92–1.00)		0.040
Differentiation								
Well to moderate	Reference							
Poor differentiation	1.41	(0.90–2.19)		0.131				
Signet ring feature	0.94	(0.71–1.24)		0.668				
Gastrectomy type								
Subtotal gastrectomy	Reference				Reference			
Total gastrectomy	1.78	(1.35–2.35)		<0.001 **	1.24	(0.71–2.16)		0.449
Lymphadenectomy type								
<D2 dissection	Reference				Reference			
D2 dissection	0.61	(0.46–0.82)		0.001 **	0.70	(0.50–0.99)		0.041 *
Stage								
Stage II	Reference				Reference			
Stage III	3.58	(2.58–4.96)		<0.001 **	2.11	(1.27–3.49)		0.004 **
Lymphovascular invasion	1.87	(1.31–2.66)		0.001 **	0.76	(0.51–1.15)		0.199
Perineural invasion	1.91	(1.42–2.57)		<0.001 **	1.24	(0.88–1.75)		0.225
site								
Upper	1.31	(0.91–1.88)		0.150	0.96	(0.52–1.78)		0.890
Middle	1.16	(0.74–1.84)		0.515	1.33	(0.81–2.19)		0.260
Lower	Reference				Reference			
More than one site	2.00	(1.27–3.14)		0.003 **	1.16	(0.54–2.49)		0.710
Stump	2.17	(1.30–3.63)		0.003 **	1.66	(0.79–3.49)		0.182
Borrmann type								
0–1	Reference				Reference			
2	1.59	(0.75–3.35)		0.224	1.52	(0.65–3.52)		0.333
3	3.21	(1.64–6.29)		0.001 **	2.59	(1.20–5.60)		0.015 *
4	7.40	(3.31–16.54)		<0.001 **	4.22	(1.56–11.41)		0.005 **
T stage								
1	Reference				Reference			
2–4	4.11	(1.53–11.07)		0.005 **	1.39	(0.46–4.24)		0.559
N stage								
0	Reference				Reference			
1–3	2.84	(1.77–4.56)		<0.001 **	2.37	(1.25–4.50)		0.008 **
Positive nodal number								
≤5	Reference				Reference			
>5	2.68	(2.04–3.53)		<0.001 **	1.68	(1.21–2.35)		0.002 **
Margin type								
0	Reference							
Radial	2.06	(1.47–2.88)		<0.001 **				
Other	2.75	(1.70–4.45)		<0.001 **				
Margin								
Negative	Reference				Reference			
Positive	2.22	(1.65–2.99)		<0.001 **	1.45	(1.05–1.99)		0.023 *

Cox’s proportional hazard regression. * *p* < 0.05, ** *p* < 0.01.

## Data Availability

The datasets generated during and analyzed during the current study are available from the corresponding author on reasonable request.

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
