# Peer review of "Assessment of the Relationship Between Positive Radial Margin and Prognosis in Patients with Gastric Adenocarcinoma"

_cancers, 2025, doi:10.3390/cancers17091463_

Round 1

Reviewer 1 Report

Comments and Suggestions for Authors

This paper addresses a retroactive cohort research. The aim of the research was to investigate how good radial margins affected the prognosis of gastric cancer patients who had a gastrectomy. Particularly relevant and focused on a crucial element of oncology research, this study addresses the prognostic value of surgical margins in gastric cancer.

Strengths:

Topic: The results for patients with stomach cancer depend much on surgical margins, hence the research looks at this important question.

Methodology: By means of a ten-year period, the retrospective cohort design allows the analysis of a large sample size of 431 patients, therefore generating a complete dataset with analytical value.

Statistical Analysis: It is admirable how several statistical techniques—including logistic regression and Cox proportional hazards regression—are used to assess the variables influencing the results, thereby improving the validity of the results and deserving for attention.

chances for development and improvement:

Areas of improvement:

Abstract: Though academic criteria call for a logical framework, the abstract lacks any of these qualities. Organising the material into many sections—including the aims and antecedents, the techniques, the outcomes, and the conclusions—helps to improve clarity and understanding.

Introduction: Although the beginning is essential for including background information, it might be improved by more exact articulation of the research topic or hypothesis within the introduction. As such, the customers would understand the exact goals of the study more holistically.

Methodology: The section on approaches may perhaps build on the criteria used to find positive radial margins. Explicit definitions and procedures are very necessary if one wants repeatability and understanding.
It is helpful to offer particular information on the confusing elements that were controlled in the study by means of data thus clarifying the management of these aspects.

Results: The section on the results has to include more precisely the data on overall survival (OS) and disease-free survival (DFS) by using extra visual aids like survival curves or diagrams. While tables are helpful, visual explanations could be better in enhancing understanding.
Moreover, the results should provide a more thorough justification of the relevance of the discoveries, especially with their therapeutic consequences.

Discussions: Including comparisons with the present corpus of literature could help the speech to be developed even further. A study of the degree to which the margin status statistics match or contradict earlier studies on the topic might provide a great deal of information.
More data should be given on the therapeutic ramifications of the results, especially in connection to the treatment plans meant for patients with positive radial margins.

Conclusion: The conclusion requires a succinct overview of the most important results, with special focus on the therapeutic relevance. A call to action for further research might help to increase the significance of the outcome even more.

Proofreading: A few little spelling mistakes and inconsistencies abound; the phrase "gastrectimy," which should be changed to "gastrectomy," should The professionalism of the writing would need thorough proofreading to raise it.

Reference: Check to be sure every reference is current and relevant. The reference part must include recent studies supporting the points of view expressed in the essay.

The study exposes significant data on the relationship between positive radial margins and the prognosis of individuals diagnosed with stomach cancer. By addressing the areas noted as needing improvement, the writers may be able to improve the clarity, rigour, and relevance of their writing. This will help them to contribute more significantly to the discipline of cancer research.

Author Response

comment 1: Abstract: Though academic criteria call for a logical framework, the abstract lacks any of these qualities. Organising the material into many sections—including the aims and antecedents, the techniques, the outcomes, and the conclusions—helps to improve clarity and understanding.

response 1: Thank you for your kind reminder. I have revised the abstract and reorganized it to ensure a more logical structure.

comment 2:  Introduction: Although the beginning is essential for including background information, it might be improved by more exact articulation of the research topic or hypothesis within the introduction. As such, the customers would understand the exact goals of the study more holistically.

response 2: Thank you. I have revised the introduction to clarify the research topic

comment 3: Methodology: The section on approaches may perhaps build on the criteria used to find positive radial margins. Explicit definitions and procedures are very necessary if one wants repeatability and understanding.
It is helpful to offer particular information on the confusing elements that were controlled in the study by means of data thus clarifying the management of these aspects.

response 3: I have added the indications for gastrectomy and provided more details regarding the surgical procedure. As mentioned, we included patients who underwent gastrectomy, and the pathological examination of the specimens was conducted to identify any cases with positive margin status, including proximal, distal, and radial margins.

comment 4: Results: The section on the results has to include more precisely the data on overall survival (OS) and disease-free survival (DFS) by using extra visual aids like survival curves or diagrams. While tables are helpful, visual explanations could be better in enhancing understanding. Moreover, the results should provide a more thorough justification of the relevance of the discoveries, especially with their therapeutic consequences.

response 4: I agree with your point. I have included the survival curves in Figure 1 and Figure 2, along with their corresponding figure legends.

comment 5: Discussions: Including comparisons with the present corpus of literature could help the speech to be developed even further. A study of the degree to which the margin status statistics match or contradict earlier studies on the topic might provide a great deal of information. More data should be given on the therapeutic ramifications of the results, especially in connection to the treatment plans meant for patients with positive radial margins.

response 5: Thank you. I have reviewed the literature in the PubMed database and found that most studies on positive radial margins and prognosis focus on esophageal and rectal cancers. There is limited data addressing this issue in gastric cancer. I have included this point in the discussion section and added the relevant references.

comment 6: Conclusion: The conclusion requires a succinct overview of the most important results, with special focus on the therapeutic relevance. A call to action for further research might help to increase the significance of the outcome even more.

response 6: Certainly. I have revised the conclusion to present the main points more clearly.

comment 7: Proofreading: A few little spelling mistakes and inconsistencies abound; the phrase "gastrectimy," which should be changed to "gastrectomy," should The professionalism of the writing would need thorough proofreading to raise it.

response 7: I sincerely apologize for the oversight. All spelling errors have been corrected. Thank you for your kind reminder.

comment 8: Reference: Check to be sure every reference is current and relevant. The reference part must include recent studies supporting the points of view expressed in the essay.

response 8: As mentioned above, there is limited literature exploring the relationship between radial margin status and prognosis in patients undergoing gastrectomy. Most existing studies focus on the association between positive proximal or distal margins and prognosis. I have reviewed the references again and added some relevant studies to support the discussion.

Thank you very much for reviewing my manuscript. I have made every effort to address the issues identified. If there are any further corrections or additions needed, please let me know, and I will do my best to improve it.

Reviewer 2 Report

Comments and Suggestions for Authors
  1. The study design and statistical methods appear to be appropriate, although the authors should clarify whether they are looking at overall margin status or radial margin status alone in the various analyses.  One wonders if it would be helpful to perform a comparison of margin status with and without the addition of the radial margin. 
  2. Grammar and syntax are acceptable, although instructions for authors have been inadvertently left in the text in two places and should be removed.
  3. The inclusion of radial margins in the margin status evaluation appears to be novel, although there is at least one published paper that includes radial margin status that should be cited and discussed here.1
  4. The conclusions are validly drawn from the findings of the study, although the authors should discuss strategies to decrease the risk of positive margins such as the use of neoadjuvant or perioperative chemotherapy.
  5. Do any published guidelines, such as NCCN, recommend evaluating radial margin status in these patients? Do the authors believe that the inclusion of radial margin status evaluation in these patients should be the standard of care?  Should guidelines be revised to include this assessment?
  6. Do the authors have any data regarding the use of neoadjuvant chemotherapy? Are there any data on patterns of recurrence?  It would be interesting to see the effect of neoadjuvant chemotherapy on both of these.
  7. Are there any data regarding the use of minimally invasive versus open surgery?
  8. The last sentence in the methods subsection of the abstract and the first sentence in the results subsection of the abstract are repetitive.
  9. In the introduction section, the authors common that gastrectomy remains one of the primary curative treatments for patients with gastric cancer. Are there any other curative treatments for stage II/III locally invasive gastric cancer?  Please consider making a stronger statement such as “margin negative resection with appropriate lymphadenectomy is considered to be a mandatory component of potentially curative treatment in patients with stage II/III locally invasive gastric cancer.”
  10. The authors indicate that all patients consented for treatment prior to confirming IRB approval of the research protocol. This would seem to go without saying and should not be necessary for a retrospective review.  Did the patients give consent to undergo treatment OR to participate in the research study?
  11. Did any of the patients have positive radial margins and positive other margins?
  12. Are there any data on clinical stage?
  1. Ma LX, Espin-Garcia O, Lim CH, Jiang DM, Sim HW, Natori A, et al. Impact of adjuvant therapy in patients with a microscopically positive margin after resection for gastric and esophageal cancers. J Gastrointest Oncol. 2020;11:356-65.

Author Response

comment 1: The study design and statistical methods appear to be appropriate, although the authors should clarify whether they are looking at overall margin status or radial margin status alone in the various analyses.  One wonders if it would be helpful to perform a comparison of margin status with and without the addition of the radial margin. 

response 1: Thank you for highlighting this important point that I had neglected to explain in the article. I have added the necessary explanation in the Results section, under 3.1.1 Patient Characteristics.

comment 2: Grammar and syntax are acceptable, although instructions for authors have been inadvertently left in the text in two places and should be removed.

response 2: Thank you for your kind reminder. I've delete it now. 

comment 3: The inclusion of radial margins in the margin status evaluation appears to be novel, although there is at least one published paper that includes radial margin status that should be cited and discussed here.

response 3: Thank you very much for providing this excellent reference. I have added it to both my discussion section(in page 11) and references(reference 18). In fact, there is limited data exploring the relationship between prognosis and radial margin in patients undergoing gastrectomy. Most of the existing data focuses on esophageal and rectal cancers.

comment 4: The conclusions are validly drawn from the findings of the study, although the authors should discuss strategies to decrease the risk of positive margins such as the use of neoadjuvant or perioperative chemotherapy.

response 4: I agree with your point. I've revise the conclusion. As the following: 
Worse overall survival and disease-free survival were linked to positive radial margin status. To improve prognosis, R1 resection should be avoided, and the use of intraoperative frozen section analysis may help verify margin status during surgery. Detailed descriptions in pathological reports are required, and future treatment plans for patients with positive radial margins should be further explored.

comment 5: Do any published guidelines, such as NCCN, recommend evaluating radial margin status in these patients? Do the authors believe that the inclusion of radial margin status evaluation in these patients should be the standard of care?  Should guidelines be revised to include this assessment?

response 5: After reviewing guidelines, including the NCCN and the Japanese Gastric Cancer Treatment Guidelines (6th edition), there appear to be no specific recommendations regarding the radial margin. Based on our data, we believe that R0 resection should involve negative margins across all sites, including the radial margin, and this should be considered the standard of care. However, as this is a retrospective study, further research is needed to determine whether the guidelines should be revised.

comment 6: Do the authors have any data regarding the use of neoadjuvant chemotherapy? Are there any data on patterns of recurrence?  It would be interesting to see the effect of neoadjuvant chemotherapy on both of these.

response 6: We did not collect data on the use of neoadjuvant chemotherapy in this cohort. Instead, we gathered information on recurrence patterns, with 68 patients experiencing peritoneal recurrence, 93 having distant metastasis, and 38 showing local recurrence. However, we did not investigate the relationship between these recurrence patterns and margin type. Thanks for giving me the idea for the next research topic. 

comment 7: Are there any data regarding the use of minimally invasive versus open surgery?

response 7: Sorry, we didn't collect the data about how much patients received minimally invasive surgery or open surgery. 

comment 8: The last sentence in the methods subsection of the abstract and the first sentence in the results subsection of the abstract are repetitive.

response 8: I am sorry for this. I've revised the mistake. 

comment 9: In the introduction section, the authors common that gastrectomy remains one of the primary curative treatments for patients with gastric cancer. Are there any other curative treatments for stage II/III locally invasive gastric cancer?  Please consider making a stronger statement such as “margin negative resection with appropriate lymphadenectomy is considered to be a mandatory component of potentially curative treatment in patients with stage II/III locally invasive gastric cancer.”

response 9: I completely agree with your point. I have revised the statement to make it clearer. Thank you very much for your valuable feedback!

comment 10: The authors indicate that all patients consented for treatment prior to confirming IRB approval of the research protocol. This would seem to go without saying and should not be necessary for a retrospective review.  Did the patients give consent to undergo treatment OR to participate in the research study?

response 10: Thank you. I've deleted the redundant sentence. 

comment 11: Did any of the patients have positive radial margins and positive other margins?

response 11: Yes. Mixed margin positive was classified as ''other'' instead of ''radial'' margin positive. In the ''radial'' margin positive group, only radial margin was positive. The proximal and distal margin were negative. I've clarified the definition in the Results section, under 3.1.1 Patient Characteristics. Thank you very much. 

comment 12: Are there any data on clinical stage?

response 12: we are sorry we didn't collect the data of clinical stage. Only pathologic stages were collected. 

Thank you very much for reviewing my manuscript. Your suggestions have been extremely helpful in revising it. I have made the necessary revisions to the best of my ability. If there are any additional revisions or additions needed, please let me know, and I will address them promptly.

Round 2

Reviewer 1 Report

Comments and Suggestions for Authors

The authors made the modifications that I required. I find it suitable for publication.